# Enhancing the Production of Pinene in *Escherichia coli* by Using a Combination of Shotgun, Product-Tolerance and I-SceI Cleavage Systems

**DOI:** 10.3390/biology11101484

**Published:** 2022-10-10

**Authors:** Ming-Yue Huang, Wei-Yang Wang, Zhen-Zhen Liang, Yu-Chen Huang, Yi Yi, Fu-Xing Niu

**Affiliations:** 1Guangxi Key Laboratory of Green Processing of Sugar Resources, Guangxi University of Science and Technology, Liuzhou 545006, China; 2Department of Basic Medicine, Guangxi University of Science and Technology, Liuzhou 545006, China

**Keywords:** tolerance, pinene, *E. coli*, limit screening method, shotgun, I-SceI

## Abstract

**Simple Summary:**

The limit screening method (LMS method) and product-tolerance engineering was proposed and applied. *AcrB*, *flgFG*, *motB* and *ndk* were found to be associated with an enhanced tolerance of *E. coli* to pinene. Using the I-SceI cleavage system, the promoters of *acrB*, *flgFG*, and *ndk* genes were replaced with P37, and pinene production was eventually increased 2.1 times. In general, a combination of shotgun, product-tolerance, and I-SceI cleavage systems provides a more convenient and efficient way of enhancing the synthesis of target products in a host strain.

**Abstract:**

Tolerance breeding through genetic engineering, sequence and omics analyses, and gene identification processes are widely used to synthesize biofuels. The majority of related mechanisms have been shown to yield endogenous genes with high expression. However, the process was time-consuming and labor-intensive, meaning there is a need to address the problems associated with the low-throughput screening method and significant time and money consumption. In this study, a combination of the limit screening method (LMS method) and product-tolerance engineering was proposed and applied. The *Escherichia coli* MG1655 genomic DNA library was constructed using the shotgun method. Then, the cultures were incubated at concentrations of 0.25%, 0.5%, 0.75% and 1.0% of pinene with different inhibitory effects. Finally, the genes *acrB*, *flgFG*, *motB* and *ndk* were found to be associated with the enhanced tolerance of *E. coli* to pinene. Using the I-SceI cleavage system, the promoters of *acrB*, *flgFG* and *ndk* genes were replaced with P37. The final strain increased the production of pinene from glucose by 2.1 times.

## 1. Introduction

The synthesis of biofuels using synthetic biology techniques is replacing conventional methods of synthesizing fuels. The low-carbon energy nature of biofuels makes them more advantageous than conventional biofuels because they have fewer effects on climate change, economic benefits and lower CO_2_ emissions than traditional fuels [1,2], such as n-butanol [3,4], n-pentanol [5], isopentanol [6], styrene [7], limonene [8], pinene [9,10], farnesene [11,12] and others. However, metabolic engineering for the sustained and reliable production of biofuels is a complex and multifaceted effort [13]. Host engineering mainly focuses on optimizing metabolic pathways, improving the supply of precursors [10] and reducing the impact of toxic substances [14]. In addition to the regulation of metabolic pathways, effectively solving the growth inhibition of biofuels in the host is another way to improve yield, and the application of tolerance engineering has become an effective means to improve such a yield [15,16,17,18]. To improve host tolerance to biofuels, some efficient physical [19], chemical mutagenesis [20] and adaptative laboratory evolution (ALE) [21,22] methods have been applied. From the analysis of the results, they mainly involve the high expression of host endogenous genes [23,24], but the screening process of endogenous genes needs to proceed through several steps of mutagenesis, including strain screening, sequencing, omics analysis, verification and regulation, which is a tedious and time-consuming process. Therefore, it is necessary to establish a general method to quickly obtain effective genes from genomic DNA and avoid the need for high-throughput screening methods.

Pinene is a naturally active monoterpene that has recently been produced as a candidate for renewable aviation fuel due to its favorable energy content, cold-weather properties, and high octane/hexadecane number [10,25]. Pinene is biosynthesized from the C5 intermediates isopentenyl diphosphate (IPP) and dimethylallyl diphosphate (DMAPP) and then condensed by geranyl diphosphate synthase (GPPS) to form geranyl diphosphate (GPP); next, it is cyclized by pinene synthase (PS) to produce pinene. These molecules can be produced by two metabolic pathways, the heterologous mevalonate (MEV) pathway (Equation (1)) and the native 1-deoxy-D-xylulose-5-phosphate (DXP) pathway in *E. coli* (Equation (2)) [26]. The pMEVIGPS plasmid contains the MEV pathway and pinene synthase for pinene production. The synthesis pathway of pinene is shown in Figure 1.
(1)$$3\mathrm{Glucose}+4\mathrm{NADPH}+12{\mathrm{NAD}}^{+}=\mathrm{Pinene}+8{\mathrm{CO}}_{2}+4{\mathrm{NADP}}^{+}+12\mathrm{NADH}$$
(2)$$2\mathrm{Glucose}+4\mathrm{ATP}+6\mathrm{NADPH}+2{\mathrm{NAD}}^{+}=\mathrm{Pinene}+2{\mathrm{CO}}_{2}+4\mathrm{ADP}+6{\mathrm{NADP}}^{+}+2\mathrm{NADH}+4\mathrm{Pi}$$

However, the production level of pinene is much lower than that of other biofuels (isoprene [27], farnesene [28], etc.) because of a higher toxic level to *E. coli*. Thus, a combination of tolerance, evolution and modular co-culture engineering had been used for pinene production [10], but the implementation of this process was a time-consuming and labor-intensive project, and the effect was also mediocre. Therefore, in this study, a combination of the limit screening method (LMS method) with tolerance engineering was proposed and applied. The LMS method was used to construct a genome library and transfer it to host cells to take advantage of the inference that the increased tolerance of host cells can increase production. Host cells containing plasmids were sub-cultured in a fuel-product-limiting environment to kill most of the cells, and the remaining cells were collected for genetic identification. This method is simple and efficient and does not need to consider the difficulty of establishing high-throughput screening methods. Moreover, it is a feasible application for increasing yield using tolerance engineering.

## 2. Materials and Methods

### 2.1. Strains, Plasmids and Primers

The strains and plasmids used in this study are shown in Table 1. The primers used in this study are shown in Appendix A.

### 2.2. Construction of E. coli Genomic Library

*E. coli* MG1655 were grown in 20 mL of LB medium at 37 °C for 12 h and harvested. Their genomic DNA was extracted via the phenol–chloroform method (collecting bacteria, broken with quartz sand and extracted by adding an equal volume of phenol and chloroform). Then, genomic DNA was divided into 200 ng/µL and partially digested with *Sau*3AI as described by Kang et al. [31]. The restriction fragments were fractionated via 0.5% agarose electrophoresis, and fragments between 1 kb and 5 kb were collected. These purified fragments (2.7 pg) were ligated with 1.4 pg of BamHI-digested, dephosphorylated pZEABP and transformed into *E. coli* DH5α. 

### 2.3. Limit Screening Method (LMS Method) Combined with Tolerance for Gene Isolation 

The *E. coli* MG1655 genomic library described above was isolated with a Qiagen extraction kit. (Tiangen Biochemical Technology, Beijing, China) Competent cells were transformed with 10 ng of DNA from the genomic library, spread on LB plates containing ampicillin (100 µg/mL) and left overnight at 37 °C. Then, all transformed daughter cells were collected and transferred to medium supplemented with 0.25% pinene, then transferred to medium with 0.5, 0.75 and 1.0% pinene concentrations at 10% inoculation every 12 h. Finally, all the culture media were collected, re-suspended with 100 µL of sterile water and coated in solid medium (LB with 1% agar).

### 2.4. DNA Sequencing and Analysis

DNA sequences for both strands were determined by primer walking at ANSHENGDA (Shuzhou, China). Sequence analysis was performed with the NCBI-BLAST function (https://blast.ncbi.nlm.nih.gov/Blast.cgi) (accessed on 8 October 2021).

### 2.5. Construction of Plasmids

The *acrB*, *flgFG*, *motB*, *pbpc* and *ndk* genes were amplified from *E. coli* and inserted into pZEABP [29] to obtain pZEA-acrB, pZEA-flgFG, pZEA-motB, pZEA-pbpc and pZEA-ndk, which were regulated by the constitutive P37 promoter of *E. coli*. In view of the preliminary target research basis [10], the I-SceI traceless cutting technique was re-constructed and optimized for use. Promoter replacement was performed by λ-red recombination and I-SceI cleavage. This method was first reported by Yu et al. [32] and modified by Wei et al. [30], and it uses the isopropyl β-D-1 thiogalactopyranoside (IPTG)-inducible *ccdB* instead of *sacB* as the counter-selectable marker to increase screening efficiency. In this study, efficient promoter replacement was the focus. In order to improve the cutting efficiency, two S sites were inserted, and the upstream and downstream homologous arms were set as the P37 promoter sequence. Therefore, the P37-S-Kan-S-P37 sequence was synthesized by Suzhou ANSHENGDA, Inc. (Suzhou, China) and ligated to pUC57 to obtain pUC57-P37-S-Kan-S-P37 (See Appendix A for sequence). For promoter replacement, primers were used (Appendix A) to amplify the upstream and downstream fragments of *acrB*, *flgFG*, *motB*, *pbpc* and *ndk*; the upstream fragments were inserted upstream of pUC57- P37-S-Kan-S-P37 by EcoRI/BamHI, and the downstream fragments were inserted downstream of pUC57- P37-S-Kan-S-P37 by SalI/HindIII. Finally, these targeted vectors of pUC-IsecI-*acrB*, pUC-IsecI-*flgFG* and pUC-IsecI-*ndk* were constructed, and targeted fragments were digested with EcoRI/HindIII and purified with a Qiagen gel extraction kit.

### 2.6. Promoter Replacement Based on I-SceI System 

The markerless promoter replacement process was carried out as reported by Wei et al. [30], E. coli MG1655 cells carrying pSIMIS were cultured in in 10 mL of LB medium at 30 °C until they were in the early log phase (OD600 = 0.4), and the expression of λred recombinant genes (gam, bet and exo) was induced by heat shock at 42 °C for 15 min. The cells were cooled to 4 °C for 10 min and centrifuged at 9000 rpm for 30 s at 4 °C. The cells were rinsed three times with frozen sterile water and resuspended in 100 µL of frozen sterile water after the salt was removed. Then, 400 ng of target fragments were electrotransformed at 1.8 kV in 1 mm gap cuvettes on a Bio-Rad MicroPulser, BTXECM-830 (BTX, Poway, CA, USA). Cells were incubated in SOC at 30 °C for 3 h and spread onto LB plates containing ampicillin (100 µg/mL) and kanamycin (50 µg/mL). PCR was performed to verify the correct replacement of the target genomic region by the markeless replacement cassette. The selection markers (Kan^R^/P37-S-Kan-S-P37) that were introduced into the genome as described above were then excised from the recombinant E. coli strains by DSB repair mediated by the I-SceI endonuclease expressed from pSIMIS [30]. Briefly, the recombinant strains were grown to OD600 = 0.4 at 30 °C in 5 mL of LB medium containing ampicillin (100 µg/mL); then, a 10 mM final concentration of arabinose was added, and cultures were heat-shocked in a shaking water bath at 42 °C for 15 min again. After three rounds of continuous culture, cultures were diluted 10-fold and plated on LB plates containing ampicillin (100 µg/mL) and arabinose (10 mM). Finally, the recombinant replicas were plated on LB plates containing ampicillin or kanamycin, and colonies were selected. The selected strains were amplified by PCR and sequenced.

### 2.7. Pinene biosynthesis in shake flasks

For pinene fermentation production, a single colony was inoculated into 5 mL of LB medium. The overnight seed culture was then inoculated into 50 mL of NB medium with a starting OD_600_ of 0.1. NB medium (pH 7.0) contained the following (g/L): glucose 10, peptone 12, yeast extract 24, MgCl_2_ 0.5 and MnCl_2_ 0.25. The main cultures were then incubated at 37 °C and 200 rpm until the OD_600_ reached 0.8. Then, the cultures were induced with 0.5 mM IPTG and overlaid with 10% dodecane to trap pinene. After induction, the cultures were incubated at 30 °C and 130 rpm for 72 h. In particular, to minimize the volatilization of pinene, the triangular bottles were sealed with a rubber sieve.

### 2.8. GC analysis

The pinene concentration in the dodecane layer was determined as previously reported [10]. Then, 500 µL dodecane layers were placed in a 1.5 mL microcentrifuge tube, centrifuged at 9000 rpm for 1 min and diluted in ethyl acetate (10×) with internal standard limonene added. The samples were analyzed via gas chromatography (GC) flame ionization detection (FID). The GC-FID instrument (Techcomp GC7900, Techcomp Ltd., Shanghai, China) was used with a TM-5 column (30 m × 0.32 mm × 0.50 μm). The inlet temperature was set to 300 °C, with the flow at 1 mL/min, oven at 50 °C for 30 s, ramp at 4 °C/min to 70 °C and ramp at 25 °C/min to 240 °C.

### 2.9. Statistical analysis

All experiments were performed in triplicate, and data were averaged and expressed as mean ± standard deviation. One-way ANOVA and Tukey’s test were used to determine significant differences using OriginPro (version 9.1) (OriginLab, Northampton, MA, USA) package. Statistical significance was defined as *p* < 0.05.

## 3. Results and Discussion

### 3.1. Isolation of Genes Conferring Increased Pinene Accumulation in E. coli Using the LMS Method

Tolerance engineering is an effective method for increasing biofuel yield. To characterize pinene tolerance, the growth of *E. coli* MG1655 was compared in different concentrations of pinene. Figure 2 shows that a 0.25% pinene concentration had a serious impact on the growth of *E. coli* MG1655. To identify genes that could improve the tolerance of *E. coli*, an *E. coli* MG1655 genomic DNA library was constructed with pZEABP (with more than 10^5^ transformants) and transferred to *E. coli* MG1655. To rapidly achieve high efficiency, an LMS method was used. After four subcultures, all transformed daughter cells were collected and transferred under solid medium with 1% pinene. However, only four single colonies were found. This may have been due to incomplete database capacity or loss during database construction, or it may have been due to the toxic effect of pinene, which makes single cells with the target gene immortal but unable to replicate, or those that have already replicated are killed again before they can function. Details of the mechanism need to be further studied. Then, single colony culture and sequencing were performed and compared with NCBI-BLAST. The clone P1 was found to contain *ndk* and part *pbpc* (non-active center); clone P2 contained *flgF* and part *flgG*; and P3 and P4 contained just one complete gene, *motB* and *acrB*, respectively (Table 2). The *ndk* gene encodes a nucleoside diphosphate kinase, which helps maintain intracellular dNTP balance and the balance between ATP and NTPs in metabolic processes and participates in signal transduction [33,34]. *flgF-flgG* encode a flagellar basal-body rod protein, which is related to cell growth and swimming [35,36]. The *motB* gene encodes a protein that enables flagellar motor rotation, a process that is also related to cell growth and swimming [37]. The *acrB* gene encodes a multidrug efflux pump resistance–nodulation–division (RND) permease AcrB which has the function of transportation. Fisher et al. [38] reported that the AcrB efflux pump has the capability of enhancing tolerance to short-chain alcohols.

Although the bacteria survived high concentrations of pinene, further cell expansion did not occur (too few colonies grew). Hence, genes that could further increase pinene production were needed for identification. The pZEABP plasmids that contained inserts of *E. coli* DNA fragments (Table 1) were co-transferred into *E. coli* MG1655 with pMEVIGPS and compared with a control strain containing the empty vector pZEABP with pMEVIGPS. As shown in Figure 3, only the genes *acrB*, *flgFG*, *motB*, and *ndk* led to an improvement in pinene production.

### 3.2. Effect of I-SceI-Induced Recombination Promoter Replacement on Pinene Production

To prepare a more dominant *E. coli* and to refer to what Wei et al. reported [30], under the regulation of their native promoters, the expression level of *E. coli* genes often cannot meet the needs of survival in extreme environments. Replacing the P37 constitutive promoter with one with a stronger ability greatly helps to improve the gene expression level (Appendix A). Genes that enhance pinene production and function needed to be replaced by promoters on chromosomes; refer to Figure 4 for the detailed process. As shown in the following schematic flow chart of promoter replacement, segment P37-*acrB* was used to replace the *acrB* native promoter and obtained *E. coli* MG1655B.

However, perhaps due to the presence of two S sites—which is double what Wei et al. reported [30]—its target efficiency did not decrease significantly in the absence of the ccdB toxic protein (80.3% identification by PCR and sequencing analysis). AcrB is an efflux pump protein associated with material transport. Dunlop et al. reported that the overexpression of AcrB from *E. coli* led to the enhancement of limonene tolerance [39]. The overexpression of tolC with ABC transporters was found to increase the amorphadiene titer by more than threefold [40]. Foo and Leong also reported evolved acrB variants with 47% and 400% increased tolerance to pinene and n-octane, respectively [41]. In our study, the replacement of the *acrB* promoter resulted in a 50.4% increase in pinene production (Figure 5C). To accurately determine whether the corresponding genes could promote an increase in pinene synthesis, pZEA-*flgFG*, pZEA-*ndk*, and pZEA-*motB* were co-transformed with pMEVIGPS after the *acrB* gene promoter was replaced. As shown in Figure 5A, *flgFG* caused a further increase of 21.3% in pinene production related to *E. coli* MG1655B (pZEABP and pMEVIGPS). Therefore, the promoter P37 fragment was inserted in front of flgF to enhance the role of flgF and flgG, and *E. coli* MG1655BF was obtained. FlgF and FlgG are flagellar proteins associated with cell growth and vitality. In our previous study, omics analysis of evolutionary pinene-tolerant bacteria showed the up-regulation of *flgFG* [24]. After that, pZEA-*ndk* and pZEA-*motB* were co-transformed with pMEVIGPS (Figure 5B). As shown in Figure 5C, *ndk* can lead to a further increase (14.7%) in pinene production, but the role of *motB* was minuscule. Thus, the promoter P37 fragment was inserted in front of *ndk* again, and *E. coli* MG1655BFN was obtained. 

Finally, *E. coli* MG1655BFN was the final strain to affect pinene production, as shown in Figure 5C, in which pinene production increased from 6.8 to 14.3 mg/L. Ndk is the nucleoside diphosphate kinase protein involved in maintaining intracellular dNTP balance. This study is the first report of Ndk in terms of enhancing pinene production because the pinene-producing plasmid pMEVIGPS (15 kb, p15A ori) could replicate more stably with this enzyme. By replacing *acrB*, *flgFG* and *ndk* promoters, pinene production was also increased, and MG1655BFN was 2.1 times higher than MG1655 (Figure 5C). To further investigate the effects of *acrB*, *flgFG*, *motB* and *ndk* on pinene synthesis, gene knockdown and gene overexpression verification should be performed, and a series of omics analyses should be further conducted.

To characterize the pinene tolerance growth of *E. coli* MG1655BFN with *E. coli* MG1655, the growth of two strains was compared after 12 h in 0%, 0.25%, 0.5%, 0.75% and 1% pinene (Figure 5D). Figure 6A,B shows the growth curves of *E. coli* MG1655BFN and *E. coli* MG1655 in the presence of 0% and 1% pinene. The growth rate of *E. coli* MG1655BFN was higher than the parental strain of *E. coli* MG1655 in 1% pinene. However, the cell densities were lower than that of the parental strain of *E. coli* MG1655 in 0% pinene. 

## 4. Conclusions

A combination of shotgun, product-tolerance, and I-SceI cleavage systems provides a more convenient and efficient way of enhancing the synthesis of target products in a host strain. This strategy provides a general idea of how production can be increased in the face of biosynthetic toxic products, but there is no good way to do so. In this study, the LMS method was used in conjunction with tolerance engineering. Through the establishment and screening of the MG1655 genomic DNA library, *acrB*, *flgFG*, *motB* and *ndk* were found to be associated with the enhanced tolerance of *E. coli* to pinene. To increase its expression level at the genome level and enhance its tolerance to increase pinene production, the I-SceI cutting system was used; the promoter fragment P37 was inserted in front of *acrB*, *flgFG* and *ndk* genes; *E. coli* MG1655BFN was constructed and used; and pinene production increased from 6.8 to 14.3 mg, a 2.1-fold increase. Moreover, the growth of this strain was significantly better than that of the parental strain at different concentrations of pinene. The application of the LMS method strategy can effectively solve the problem of the efficient screening of target genes from a gene library in the process of biosynthesis. The application of the I-SceI cleavage and promoter replacement strategy provides a method to enhance the traceless replacement of endogenous gene expression.

## Figures and Tables

**Figure 1 biology-11-01484-f001:**
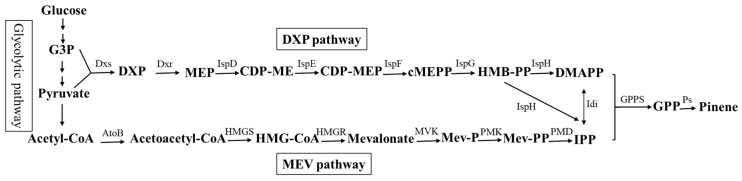
Schematic diagram of the synthesis pathway of pinene. G3P: glyceraldehyde 3-phosphate; Dxs: DXP synthase; DXP: 1-deoxy-D-xylulose-5-phosphate; Dxr: 1-deoxy-D-xylulose 5-phosphate reductoisomerase; MEP: 2-C-methyl-D-erythritol-4-phosphate; IspD: 2-C-methyl-D-erythritol 4-phosphate cytidylyltransferase; CDP-ME: 4-diphosphocytidyl-2-C-methyl-D-erythritol; IspE: 4-diphosphocytidyl-2-C-methyl-D-erythritol kinase; CDP-MEP: 4-diphosphocytidyl-2-Cmethyl-D-erythritol-2-phosphate; IspF: 2-C-methyl-D-erythritol 2,4-cyclodiphosphate synthase; cMEPP: 2-C-methyl-D-erythritol-2,4-cyclodiphosphate; IspG: (E)-4-hydroxy-3-methylbut-2-enyl-diphosphate synthase; HMBPP: 4-hydroxy-3-mehtyl-butenyl 1-diphosphate; IspH: 4-hydroxy-3-methylbut-2-en-1-yldiphosphate reductase; DMAPP: dimethylallyl diphosphate; AtoB: acetoacetyl-CoA thiolase; HMGS: HMG-CoA synthase; HMG-CoA: 3-hydroxy-3-methyl-glutaryl-CoA; HMGR: HMG-CoA reductase; MVK: mevalonate kinase; Mev-P: mevalonate 5-phosphate; PMK: phosphomevalonate kinase; Mev-PP: mevalonate diphosphate; PMD: mevalonate diphosphate decarboxylase; IPP: isopentenyl diphosphate; GPPS: geranyl diphosphate synthase; GPP: geranyl diphosphate. Idi: isopentenyl-diphosphate isomerase; PS: pinene synthase.

**Figure 2 biology-11-01484-f002:**
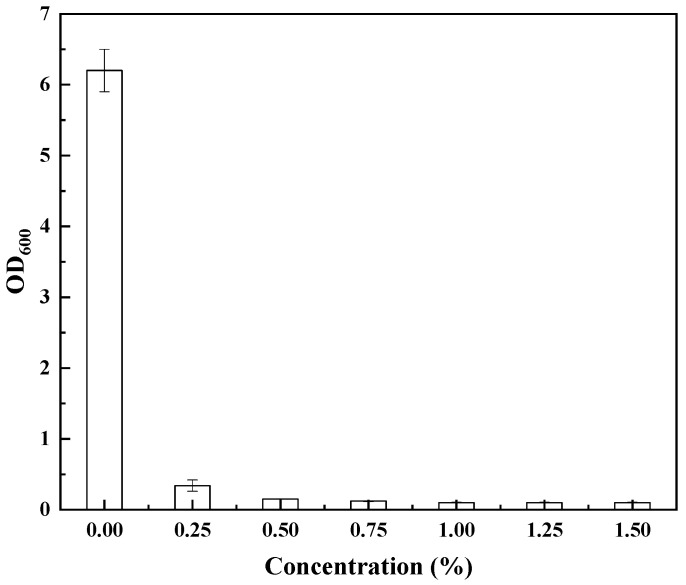
Effect of varying pinene concentrations on the growth of *E. coli*.

**Figure 3 biology-11-01484-f003:**
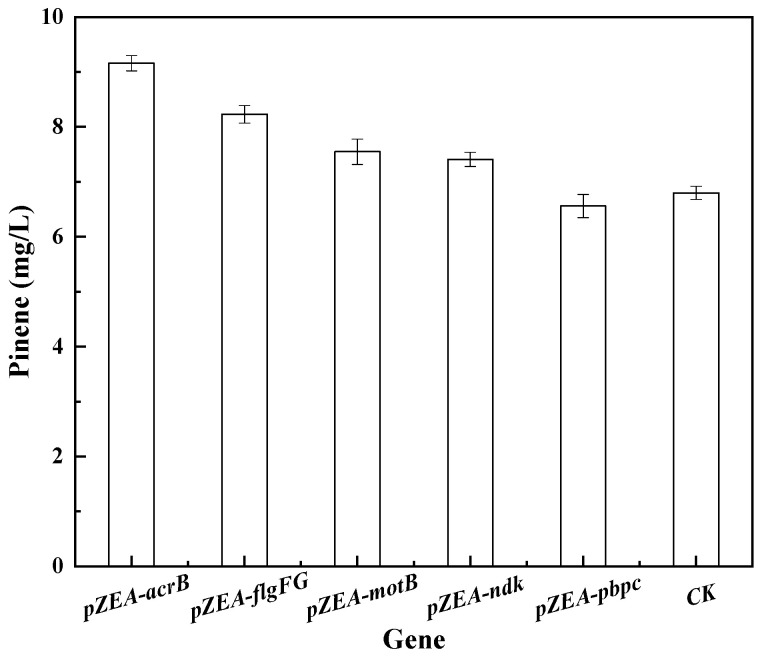
Effect of overexpression of selected genes on pinene production. Expression of plasmid in *E. coli* MG1655 (pMEVIGPS). *E. coli* MG1655 (pZEABP and pMEVIGPS) was set as the control (CK).

**Figure 4 biology-11-01484-f004:**
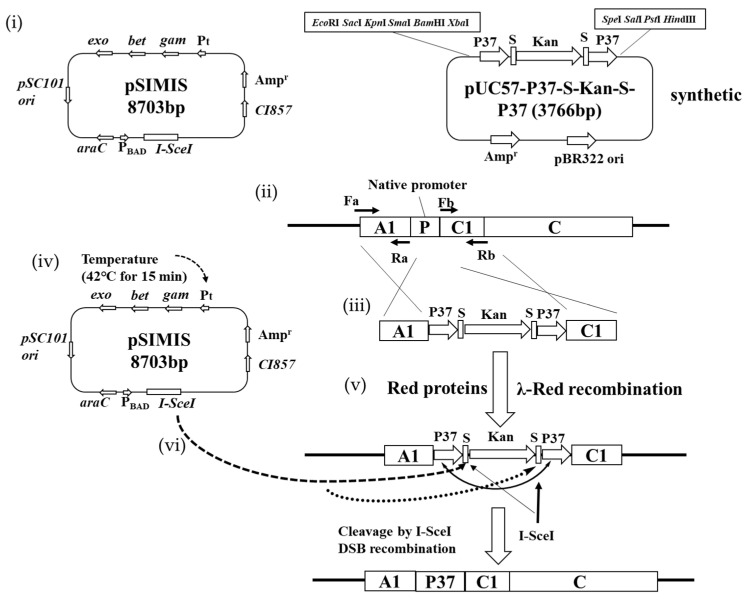
Description of rapid markerless promoter replacement with pSIMIS. (i) Plasmids used for λ-Red recombination and I-SceI cleavage. (ii) To replace the native promoter of target gene C, the left and right homologous arms were separately amplified by PCR using primers F_a_/R_a_ or F_b_/R_b_, and then cloned into the left or right MCS of pUC57-P37-S-Kan-S-P37. (iii) A linear DNA cassette containing the left arm A1, P37-S-Kan-S-P37 and right arm C1 was obtained by restriction of enzymatical digestion or PCR. (iv) The linear DNA cassettes were electroported into *E. coli* cells containing pSIMIS. (v) With help of the λ-Red proteins, the cassette could replace a target genomic segment after temperature induction. (vi) The I-SceI endonuclease in pSIMIS was induced by 10 mM arabinose to remove the introduced selection marker, and the chromosome was cleaved at the I-SceI recognition site (S). Then, the double-stranded break (DSB)-mediated intramolecular recombination between the two P37 led to the removal of the selection marker, yielding a clean, markerless promoter replacement strain.

**Figure 5 biology-11-01484-f005:**
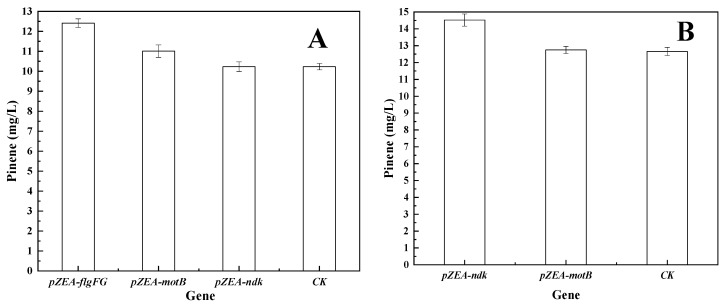
Effect of targeted strains on pinene production and growth. (**A**) Expression of plasmid in *E. coli* MG1655B (pMEVIGPS). *E. coli* MG1655B (pZEABP and pMEVIGPS) was set as the control (CK). (**B**) Expression of plasmid in *E. coli* MG1655BF (pMEVIGPS). *E. coli* MG1655BF (pMEVIGPS and pZEABP) was set as the control (CK). (**C**) Pinene production by the targeted strains harboring pMEVIGPS. (**D**) Growth of the selected strains (without plasmids) with different pinene addition after 12 h.

**Figure 6 biology-11-01484-f006:**
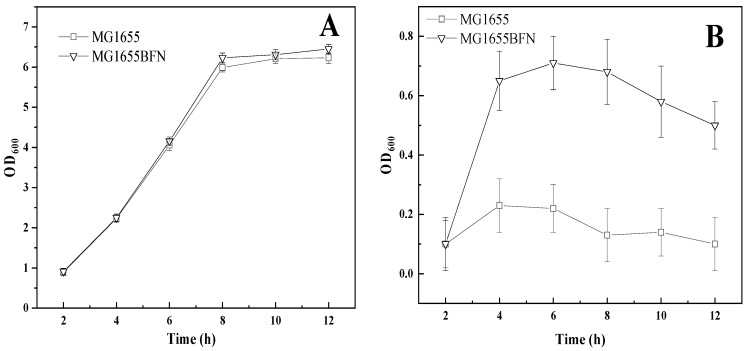
Growth of the *E. coli* MG1655BFN compared with *E. coli* MG1655 in the presence of 0% pinene (**A**) and 1% pinene (**B**). (-□-) *E. coli* MG1655; (-∇-) *E. coli* MG1655BFN.

**Table 1 biology-11-01484-t001:** Strains and plasmids used in this study.

Strains/Plasmids	Description	Source/Purpose
Strain		
*E. coli* DH5α	supE44 Δ(lacZYA-argF) U169 (Φ80lacZ ΔM15) hsdR17 recA endA1 gyrA96 thi-1 relA1	Invitrogen
*E. coli* MG1655	F- λ- ilvG- rfb-50 rph-1	ATCC#700926
*E. coli* MG1655B	*E. coli* MG1655, P37-*acrB*	This study
*E. coli* MG1655BF	*E. coli* MG1655B, P37-*flgFG*	This study
*E. coli* MG1655BFN	*E. coli* MG1655BF, P37-*ndk*	This study
**Plasmid**		
pMEVIGPS	pBbA5c-MTSAe-T1f-MBI(f)-T1002i-trGPPSA.grandis-PSA.grandi coding for MEV pathway enzymes to produce pinene from glucose in *E. coli*, p15A ori, PlacUV5 promoter, Cm^R^	[10]
pZEABP	Expression vector, P37 promoter, pBR322 ori, Amp^R^, BglBrick, ePathBrick contains four isocaudamer (AvrII, NheI, SpeI, and XbaI)	[29]
pZEA-*acrB*	pZEA*BP contains *acrB* of *E. coli*, P37 promoter, pBR322 ori, Amp^R^	This study
pZEA-*flgFG*	pZEA*BP contains *flgFG* of *E. coli*, P37 promoter, pBR322 ori, Amp^R^	This study
pZEA-*motB*	pZEA*BP contains *motB* of *E. coli*, P37 promoter, pBR322 ori, Amp^R^	This study
pZEA-*ndk*	pZEA*BP contains *ndk* of *E. coli*, P37 promoter, pBR322 ori, Amp^R^	This study
pZEA-*pbpc*	pZEA*BP contains *pbpc* of *E. coli*, P37 promoter, pBR322 ori, Amp^R^	This study
pUC57	T vector, pBR322 ori, Amp^R^	Addgene
pUC57-P37-S-Kan-S-P37	pUC57 contains P37-S-Kan-S-P37 cassette	This study
pUC-IsecI-*acrB*	pUC57-P37-S-Kan-S-P37 containing upstream and downstream homologous arms for promoter replacement of *acrB*	This study
pUC-IsecI-*flgFG*	pUC57-P37-S-Kan-S-P37 containing upstream and downstream homologous arms for promoter replacement of flgF	This study
pUC-IsecI-*ndk*	pUC57-P37-S-Kan-S-P37 containing upstream and downstream homologous arms for promoter replacement of *ndk*	This study
pSIMIS	pSC101 repliconts PL-*gam-bet-exo cI*857, arabinose-inducible I-SceI endonuclease gene, Amp^R^	[30]

**Table 2 biology-11-01484-t002:** Identification of the genes in the clones by using BLAST searches.

Positive Clone	Gene	Protein	Reference
P1	*pbpc-ndk*	peptidoglycan glycosyltransferase PbpC, nucleoside diphosphate kinase	[33,34]
P2	*flgF-flgG*	flagellar basal-body rod protein FlgF, flagellar basal-body rod protein FlgG	[35,36]
P3	*motB*	protein that enables flagellar motor rotation	[37]
P4	*acrB*	multidrug efflux pump RND permease AcrB	[38]

All the comparisons were 100% similar to BLAST searches.

## Data Availability

Not applicable.

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
