# Peer review of "Enhancing the Production of Pinene in Escherichia coli by Using a Combination of Shotgun, Product-Tolerance and I-SceI Cleavage Systems"

_biology, 2022, doi:10.3390/biology11101484_

Round 1
Reviewer 1 Report (Previous Reviewer 2)
This manuscript has been revised, and some problems have been solved. However, there are still some details should be further revised.
1. The authors still got mixed up with the use of yield and titer, and the use of "yield" in line 269 was wrong. please modify it carefully.
2. The authors have not demonstrated directly that the expression levels of target genes with replacement of P37 promoter were higher than the native promoter. Therefore, it is recommended that the authors should apply the qPCR to demonstrate that the improvement of pinene production was resulted from the overexpression of target genes by replacing with P37 promoter directly.
3. Line 212, the "Table. 2" was wrong, please modify it.
Author Response
Thank you very much for your suggestion. Please refer to the attachment for details.

Reviewer 2 Report (Previous Reviewer 3)
Can be published in the present form.
Author Response
Thank you very much.
Reviewer 3 Report (Previous Reviewer 4)
The resubmitted manuscript satisfied my previous requests.
A minor thing is the consistency of the figure style. The "A, B, C, and D" of Figure 5 are on the top right, which is different from Figure 6. For consistency, I suggest having the "A, B, C, and D" outside ALL figures' frames. This is not required, but up to the author's decision.
Other than that, the manuscript is ready for publication.
Author Response
Thank you very much for your suggestion. Please refer to the attachment for details.

Round 2
Reviewer 1 Report (Previous Reviewer 2)
This work offers great potential and an intertwined approach to study the mechanisms involved in the tolerance of toxic compounds, which may be useful for biosynthetic applications. In addition, the manuscript was carefully revised and well written for publication in the journal Biology.
This manuscript is a resubmission of an earlier submission. The following is a list of the peer review reports and author responses from that submission.
Round 1
Reviewer 1 Report
See file included

Reviewer 2 Report
The manuscript was revised and improved based on the comments, and the revised paper almost meets the publication standards. However, there are still some formatting issues that need to be corrected. For example, line 72 is missing punctuation. Authors should double-check the complete manuscript for accuracy.
Reviewer 3 Report
General comments
Huang and co-researchers report the enhancement of pinene synthesis in Escherichia coli through different genetic engineering strategies. This topic is not new, many published papers that already showed high pinene production in E. coli which is better than this present manuscript. What is new about this paper are the methods they used to increase the pinene production in E. coli but this can be affirmed only if they include a comparative literature analysis in a form of a table showing yield, titer, and productivity of pinene as well as the engineering strategies used to enhance pinene production from the previous papers versus the present data. There are also many significant issues in this manuscript that need to be addressed. First, some factual claims in the manuscript are not duly supported by the literature – additional citations are required. Second, most of the figures and figure captions are not presented well thereby affecting the overall presentation of the paper. Third, the summary and abstract need to be revised to help the readers easily understand the content of the manuscript. Fourth, there are many typographical errors in this paper as well as statements that cannot be easily understood. Finally, the authors should take note of the clarity of their main text, presentation and interpretation, and discussion. This paper is not suitable for publication in its current form. The authors need to address the following issues so their paper may be considered for publication in the future:
Specific comments
Title
· The word “tolerance system” is unusual in metabolic engineering topics. The authors should choose a more appropriate alternative word that is acceptable to the scientific community.
Summary and abstract
· The summary and abstract of the present manuscript are almost the same. The authors should present in the summary section the primary goal and the key findings of the study instead of repeating what is already written in the abstract. It is also not discussed properly why the combination of a shot-gun, tolerance, and I-SceI cleavage systems provides a more convenient and efficient way of enhancing the synthesis of target products in a host strain.
· The last sentence of the summary section (Lines 18 – 21) is extremely vague. Please paraphrase it to a more clear one.
· Also, paraphrase the 3rd sentence of the abstract section (Lines 24 - 26) so it can be easily understood.
· Please also indicate in your summary and abstract the type of tolerance you target to enhance in this study for example pH tolerance, heat tolerance, end product tolerance, a etc.
· Please also check the genes that are not italicized.
· Also include the starting substrate used to produce pinene. It should always be emphasized in the summary and abstract.
Keywords
· Include pinene as one of the keywords.
Methods
· Please indicate (Lines 82-84) if the collected DNA fragments were purified.
· Specify the solid medium used in this study. Apply it also to other sections of the manuscript.
· Please define the medium used in this study. Indicate the components.
· The 2.4 section of the methods “DNA sequencing and analysis” was not discussed thoroughly. Please discuss how sequence analysis was done using NCBI-BLAST.
· Line 99-100 “by inserting ABP and inserted into pZEABP to obtain, which were…” is an incomplete sentence. Please correct this one.
· How can you use frozen sterile water for washing? Please double-check this one.
Results and Discussion
· Please check the mentioned comment for the abstract and summary about tolerance engineering. Indicate the kind of tolerance that was enhanced in this study.
· “Figure 1 shows that 0.15% pinene concentration had a serious effect on the growth of E. coli”.The 0.15% is not presented in figure 1. Please double-check.
· Changed transferred to transformed (Line 170).
· The legend for Figure 1 is vague. You may want to change it to “Effect of varying pinene concentrations on the growth of E. coli”
· Figure 2 should be presented as a figure (drawing or illustration) instead of a flowchart.
· The legend for figure 3 should be categorized. First, define all the proteins involved and then the intermediates.
· Figure 6A-D should be put in one place together to avoid confusion.
· There is a typo in line 288. It should be figure 6C, not 6A.
· Figure 7A showed no difference between the two strains. Please revised the discussion about this one.
Reviewer 4 Report
The authors combine LMS method and tolerance engineering to optimize pinene production in bacteria. They first identified acrB, flgFG, and ndk genes that were associated with improved pinene production, then replaced their native promoters with P37, showing improved pinene production. I am convinced that the content fits the scope of Biology Journal, but the manuscript needs substantial revisions before publication.
Major concerns:
1. line 167: it should be 0.25% according to figure 1.
2. line 170: Are 10^5 transformants sufficient? What is the coverage of the genome? The author can use statistics to estimate how many transformants are sufficient to cover the genome with the specific gel-extracted DNA fragments.
3. line 263: How did you get 80.3%?
4. line 269: Figure was mislabelled.
5. Why replacing the native promoters will increase pinene production? What are the expression patterns (log phase and stationary phase) of the native promoters? What are the expression levels after replacing their promoters with P37? Are they lower or higher? Checking these aspects may gain insights into their mechanism of increasing pinene yield.
6. There is statistical analysis in the method section but no statistical indications (error bar and significance) of any figures. Are the error bars standard deviations or standard errors?
Minor concern:
1. Figure 2: transferred -> transformed